



# Reliability-based strength modification factor for seismic design spectra considering structural degradation

Sonia E. Ruiz[1], Ali Rodríguez-Castellanos[1], Edén Bojórquez[2], Miguel A. Orellana[1] and Alfredo Reyes-Salazar[2]

[1]Instituto de Ingenieria, Universidad Nacional Autonoma de Mexico, Ciudad Universitaria Coyoacan, C.P. 04510 CDMX, Mexico.
[2]Facultad de Ingeniería, Universidad Autonoma de Sinaloa, Calzada de las Americas y B. Universitarios s/n, C.P. 80040 Culiacan, Sinaloa, Mexico

*Correspondence to*: Sonia E. Ruiz (sruizg@iingen.unam.mx) and Edén Bojórquez (eden@uas.edu.mx)

**Abstract.** For earthquake resistant design, structural degradation is considered using traditional strength modification factors, which are obtained via the ratio of the nonlinear seismic response of degrading and non-degrading structural single degree of freedom (SDOF) systems. In this paper, with the aim to avoid the nonlinear seismic response to compute strength modification factors, a methodology based on probabilistic seismic hazard analyses (PSHA) is proposed in order to obtain strength modification factors of design spectra which consider structural degradation through the spectral-shape intensity measure $I_{Np}$. PSHA using $I_{Np}$ to account for structural degradation, and $Sa(T_1)$ which represents the spectral acceleration associated with the fundamental period and does not consider such degradation, are performed. The ratio of the uniform hazard spectra in terms of $I_{Np}$ and $Sa(T_1)$, that represent the response of degrading and non-degrading systems, provide new strength modification factors without the need to develop nonlinear time history analysis. A mathematical expression is fitted to the ratios that correspond to systems located in different soil types. The expression is validated by comparing the results with those derived from nonlinear time-history analyses of structural systems.

## 1 Introduction

Structures subjected to cyclic loading induced by intense ground motions can exhibit stiffness and/or strength degradation due to the inelastic nonlinear behavior of their structural elements, which can give place to lengthening of the structural fundamental vibration period $T_1$. The effect of such lengthening can be beneficial for structures whose fundamental period is in the descendant branch of the acceleration response spectrum and their higher vibration modes have little influence on the structural response. On the contrary, the effect can be detrimental, for structures whose vibration period is in the ascendant branch of the response spectrum. In the latter case, the effect of "structural softening" can have severe consequences because the structure undergoes to seismic loading greater than that assumed for its design (Akkar et al., 2004; Chenouda and Ashraf Ayoub, 2008; Chopra and Chintanapakdee, 2004; Terán-Gilmore and Espinosa Johnson, 2008). For example, during the Guerrero-Michoacán September 19, 1985 Mexican earthquake, many mid-rise buildings (5- to 10-story buildings) with



$T_1$=0.7s-1.2s approximately, located in soft soil of Mexico City, which has a vibration period around 2s, suffered severe structural damage (including collapse) because of the degrading structural effect (Montiel and Ruiz, 2007).

Seismic design guidelines for building structures recommend modifying the response-spectra ordinates by a series of factors in order to include relevant structural behavior that affects the structural response. Those factors are related, for example, to seismic behavior, structural over-strength, structural irregularity, degrading behavior, etc. A common practice to derive those modification factors is by means of the ratio between specific response spectra of single-degree-of-freedom (SDOF) systems. Indeed, most current seismic codes provisions implement simplified analyses based on these ratios. For example, the Federal

Emergency Management Agency (FEMA) introduced the so-called Coefficient Method (FEMA-273, 1997; FEMA-356, 2000), which consists of multiplying the elastic design spectrum by several coefficients. One of them takes into account the hysteretic structural degrading behavior. More recently, FEMA-440 (2005) presented some improvements to current nonlinear analysis procedures. Accordingly, the Coefficient Method suffered slight adjustments, where the coefficient that incorporates the effect of degrading structural behavior was updated. At present, the simplified nonlinear approach is available in FEMA

P-58-1 (2012) methodology. Another example is the Manual for Civil Structures Design (MCSD, 2008, 2015), developed by the Federal Commission of Electricity of Mexico, which specifies a degrading factor that increases or decreases the design spectral ordinates, due to structural deterioration.

The hysteretic degrading behavior is particularly severe for structures located in soft soil, like that in the lake bed zone of Mexico City, where there is a high-density population, and the site effects make it susceptible to severe earthquake damage

(Singh et al., 1988, 2018). In spite of that, the current Mexico City Building Code (MCBC, 2017), does not specify any structural degrading factor.

This study is aiming to propose a methodology for obtaining a mathematical expression corresponding to a structural degrading factor for seismic design of structures that exhibit period lengthening. The expression is a function of both the structural period and the dominant period of the soil. The methodology can be applied to any high seismic hazard region of the world.

**2 Methodology proposed**

In first place, it is necessary to perform PSHAs corresponding to a firm ground site, and then, to soft soil sites located in the seismic area of interest. PSHAs are associated with $Sa(T_1)$ and alternatively to $I_{Np}$ intensity measures; where $Sa(T_1)$ represents the spectral acceleration at the fundamental period of a structure, and $I_{Np}$ is an intensity measure that accounts for the period lengthening due to structural degrading behavior ($I_{Np}$ is defined below). Although $Sa(T_1)$ is the most used ground-motion

intensity measure (IM) around the world for PSHAs, it has some limitations. For example, it does not consider the effect of period lengthening of the structure due to its nonlinear behavior and mechanical properties degradation (Baker and Cornell, 2005; Bojorquez et al., 2008; Bojórquez et al., 2017a; Bojórquez and Iervolino, 2011; Kostinakis et al., 2018; Mehanny and Cordova, 2004; Shome et al., 1988; Tothong and Luco, 2007).

In second place, uniform hazard spectra (UHS) of $I_{Np}$ and $Sa(T_1)$, which represent the response of degrading and non-degrading

systems, respectively, are obtained. The UHSs are computed for several seismic recording stations located in different soil



conditions. Subsequently, the effect of the structural degradation on the response of SDOF systems is characterized by the ratio between the uniform hazard spectra: $I_{Np}/Sa(T_1)$.

Finally, a mathematical expression is adjusted to the spectral ratios. In order to verify that the mathematical expression leads to reasonable results it is convenient to compare these with those obtained with other expressions found in the literature.

In what follows, a description of the methodology is presented (see Fig. 1):

•In first place, PSHAs are carried out for the firm ground site of interest, corresponding to $Sa(T_1)$ and, alternatively, to $I_{Np}$. With the purpose of performing the analyses, the seismic tectonic zones that contribute to the seismic hazard of the site, are identified.

•Then, the probability distribution for earthquake magnitude and source-to-site distance are assumed. Additionally, it is

necessary to define adequate ground motion prediction equations (GMPEs).

•With the total probability theorem and the information previously defined, the mean annual rates of exceedance (seismic hazard curves) corresponding to the site located in firm ground, are obtained.

•Once the hazard curves for firm ground are available, the mean annual rates of exceedance of seismic recording stations located in different soil types of the seismic area of interest, are estimated (using a technique described in the following

sections). The stations are grouped in different zones, which depend on the dominant period of the soil, $T_s$.

•For each recording station site, UHS associated with a given return period, are computed for $Sa(T_1)$, and alternatively, for $I_{Np}$.

•Next, the spectral ratios $I_{Np}/Sa(T_1)$ are estimated for each site. $I_{Np}/Sa(T_1)$ represents the ratio of strength demands between systems with degrading and systems with non-degrading structural behavior.

•Finally, a simplified mathematical expression is adjusted to the spectral ratios $I_{Np}/Sa(T_1)$. The expression contains parameters

that depend on the zone of interest.

•The results of the expression proposed are compared with those obtained from other expressions found in the literature, which were obtained from time history analyses.

For illustrative purpose, in the following sections, the methodology proposed above is applied in order to find mathematical expressions of structural degrading factors of the design spectra specified in MCBC however, the approach can be applied to

any seismic region in the world.

## 3 Probabilistic seismic hazard analysis (PSHA)

### 3.1 Earthquake sources

The evaluation of a probabilistic seismic hazard analysis for a particular site requires identifying all possible earthquake sources capable of producing a significant seismic event. For this purpose, Zúñiga et al., (2017) proposed a seismic regionalization for

Mexico, which is used in the present study. Fig. 2a shows the shallow-depth seismic zones where interplate earthquakes occur due to the subduction of the Rivera and Cocos plates (SUB1, SUB2, SUB3 and SUB4). Fig. 2b illustrates the intermediate-depth seismic zones. This region corresponds to intraslab events that take place inside the subducted Rivera and Cocos plates





below south-central Mexico (IN1 to IN3). Additionally, Fig. 2c displays the seismic zones for characteristic seismic events (C1 to C14) proposed by Ordaz and Reyes (1999). Seismic zones in Fig. 2c are also included in the present study to compute

PSHA.

### 3.2 Magnitude probability distribution

Earthquake sources are capable of producing different earthquake sizes. Therefore, it is crucial to define the probability distribution of the earthquake magnitudes and corresponding rates of occurrence for each source. In this sense, the distribution of earthquake sizes is commonly described by the bounded Gutenberg-Richter recurrence law (Eq. 1).

$$\lambda_m = \nu \frac{\exp\left[-\beta\left(M_w - M_{min}\right)\right] - \exp\left[-\beta\left(M_{max} - M_{min}\right)\right]}{1 - \exp\left[-\beta\left(M_{max} - M_{min}\right)\right]} \tag{1}$$

where $\lambda_m$ is the mean annual rate of exceedance for earthquakes between a minimum magnitude $M_{min}$ and a maximum magnitude $M_{max}$, $\nu = \exp(\alpha - \beta M_{min})$ is the mean annual number of earthquakes of magnitude $M_w \geq M_{min}$, where $\alpha = 2.303p$ and $\beta = 2.303q$. The values of $p$ and $q$ are indicated in Figs. 2a and 2b, according to Zúñiga et al. (2017).

For the seismic sources related to characteristic earthquakes (Fig. 2c), the bounded Gutenberg-Richter recurrence law does not

accurately describe the magnitude exceedance rates. Accordingly, for $M_w > 7$, we employ a Gaussian probability distribution function (pdf) of magnitudes to account for the characteristic events in the Mexican subduction zones (see Eq. 2) (Ordaz and Reyes, 1999).

$$\lambda_m = \nu_7 \left[1 - \Phi\left(\frac{M_w - E_{M_w}}{\sigma_{M_w}}\right)\right] \tag{2}$$

where $\nu_7$ is the mean annual number of earthquakes of magnitude $M_w > 7$; $E_{Mw}$ and $\sigma_{Mw}$ are the mean and standard deviation of

the magnitude, respectively, and $\Phi(.)$ is the normal distribution function. The corresponding parameters to evaluate the distribution are shown in Fig. 2c.

The present study assumes $M_{min} = 4.5$ and $M_{max} = 6.9$ for the interplate shallow-depth seismic zones SUB1, SUB2 and SUB3 (see Fig. 2a). In contrast, $M_{min} = 4.5$ and $M_{max} = 7.2$, 7.8 and 7.9 are assumed for IN1, IN2, and IN3, respectively (intermediate-depth seismic zones, Fig. 2b). Finally, $M_{min} = 7.0$ and $M_{max} = 8.1$ are assumed for the fourteen earthquake sources shown in Fig. 2c.

### 3.3 Source-to-distance distribution

Once the earthquake magnitudes distribution is established, the pdf of distances from the earthquake location to the site of interest must be characterized. A uniform pdf is generally assigned to any point in the seismic zone (McGuire, 1995; Steven L. Kramer, 1996). Since the area sources, on which earthquakes can occur, are well-delimited (Figs. 2a, 2b, and 2c), it is straightforward to determine the source-to-distance distribution.





## 3.4 Ground motion prediction equations

Ground-motion prediction equations are fundamental for PSHA. They are commonly developed to predict the peak ground acceleration, PGA, or the spectral acceleration, $Sa(T_1)$. Unfortunately, attenuation models have not yet devised to provide $I_{Np}$ as a function of the vibration period (as it is done with existing GMPEs); however, with GMPEs for $Sa(T_1)$ currently available, it is possible to perform PSHA using $I_{Np}$. Here we employ the GMPEs proposed by Reyes et al., (2002) and Jaimes et al., (2015) for interplate and intraslab events, respectively. They were developed using exclusively accelerometric data recorded in Ciudad Universitaria station (CU), which corresponds to firm ground of Mexico City.

## 3.5 Seismic hazard curves

The final product of a PSHA can be expressed in different forms. Seismic hazard curves are used frequently to represent the seismic hazard. They indicate the annual rate of exceeding a variety of intensity levels of a ground motion parameter at a site of interest. The procedure to compute a ground-motion hazard curve is based on the total probability theorem (Cornell, 1968; Esteva, 1968; McGuire, 1995; Steven L. Kramer, 1996; Baker, 2008).

## 4 $I_{Np}$ Intensity measure

In order to overcome the limitations of traditional IMs (e.g., PGA, $Sa(T_1)$), advanced seismic IMs have been proposed. Some researchers suggest using vector-valued ground motion IMs. By including two or more representative parameters of the ground motion, accurate evaluations of seismic performance can be achieved (Baker and Cornell, 2005; Bojorquez et al., 2008; Bojórquez et al., 2017a; Bojórquez and Iervolino, 2011; Kostinakis et al., 2018; Mehanny and Cordova, 2004; Tothong and Luco, 2007). Accordingly, Bojórquez et al., (2008) developed the vector-valued intensity measure <$Sa(T_1)$, $Np$>, where $Np$ is a parameter proxy for the spectral shape, this IM is an advancement in predicting the seismic response in comparison with other IMs. However, the evaluation of PSHA using vector-valued IMs is a complicated and impractical task; therefore, Bojórquez and Iervolino (2011), introduced a scalar IM based on $Sa(T_1)$ and $Np$, called $I_{Np}$, both $Np$ scalar and vector-valued intensity measures have been effectively used (Bojórquez et al., 2012, 2017b).

Accordingly, Buratti (2012), made an exhaustive comparison of the most influential scalar IMs available in the literature respect to efficiency and sufficiency. The study concluded that the most effective intensity measure was $I_{Np}$. Additionally, De Biasio et al., (2014), based on a comparative study of structures with nonlinear behavior, showed the good performance of $I_{Np}$ to predict maximum interstory drift and maximum ductility demands. Moreover, Modica and Stafford (2014) using <$Sa(T_1)$, $Np$>, estimated the fragility and performance of buildings with higher efficiency respect to different IMs. In this context, Minas and Galassos (2019) showed the advantages of $I_{Np}$ comparing $Sa(T_1)$ fragility curves, for different damage states. Additionally, Yakhchalian et al., (Yakhchalian et al., 2015) demonstrated the efficiency of the parameter $Np$. They showed that the parameter $Np$ works appropriately, particularly in performance levels related to moderate levels of nonlinearity. Similarly, Kostinakis et al., (2016), proved the adequate efficiency of $I_{Np}$ to reduce the uncertainty in the prediction of the response of reinforced


concrete buildings. In addition, Jamshidiha et al., (2018), examined the ability of different IMs for predicting the seismic collapse capacity of steel resisting moment frames with fluid viscous dampers. They concluded that the scalar IM that resulted from the combination with the parameter $Np$ was most efficient.

Based on the literature mentioned above, the authors of the present study concluded that $I_{Np}$ is a promising tool to perform PSHA.

### 4.1 Methodology to perform a PSHA using $I_{Np}$

In this section a methodology to perform PSHA using $I_{Np}$ is proposed. In first place, $I_{Np}$ is defined as follows (Bojórquez and Iervolino, 2011):

$$I_{Np} = Sa(T_1) \cdot N_P{}^{\alpha} \tag{3}$$

$$N_P = \frac{Sa_{avg}(T_1...T_N)}{Sa(T_1)} \tag{4}$$

where $I_{Np}$ is the scalar intensity measure, $\alpha$ is a parameter that should be calibrated according to the structure and the earthquake demand parameter selected (in this study $\alpha=0.5$ is adopted, as recommended in Bojórquez and Iervolino, 2011); $Sa_{avg}(T_1...T_N)$ is the geometric mean of the spectral acceleration at $N$ numbers of structural vibration periods considered. $Sa_{avg}(T_1... T_N)$ takes into account the vibration period lengthening due to structural damage, and is expressed as:

$$Sa_{avg}(T_1...T_N) = \left( \prod_{i=1}^{N} Sa(T_i) \right)^{1/N} \tag{5}$$

Substituting Eq. (4) and Eq. (5) into Eq. (3), applying the natural logarithm, it results:

$$\ln(I_{Np}) = (1-\alpha)\ln[Sa(T_1)] + \frac{\alpha}{N}\sum_{i=1}^{N}\ln[Sa(T_i)] \tag{6}$$

Then, the expected value and the variance of $\ln(I_{Np})$ can be expressed as in Eq. (7) and Eq. (8), respectively.

$$E[\ln(I_{Np})] = (1-\alpha)E\{\ln[Sa(T_1)]\} + \frac{\alpha}{N}\sum_{i=1}^{N}E\{\ln[Sa(T_i)]\} \tag{7}$$

$$Var[\ln(I_{Np})] = \alpha^2 Var\{\ln[Sa_{avg}(T_1...T_N)]\} + (1-\alpha)^2 Var\{\ln[Sa(T_1)]\} \tag{8}$$

$$+ 2\alpha(1-\alpha)\rho_{\ln[Sa_{avg}(T_1...T_N)],\ln[Sa(T_1)]}\sigma_{\ln[Sa_{avg}(T_1...T_N)]}\sigma_{\ln[Sa(T_1)]}$$

The values of $\ln[Sa(T_i)]$ are obtained from existing attenuation models (e.g., the GMPEs described in section 2.4). On the other hand, $\ln[Sa(T_i)]$ terms are commonly assumed to have joint Gaussian pdf; consequently, the summation has also Gaussian



distribution. Therefore, the variance $Var\{\ln[Sa_{avg}(T_1\dots T_N)]\}$ and the correlation coefficient $\rho\ln[Sa_{avg}(T_1\dots T_N)]\ln[Sa(T_i)]$ can
be obtained by Equations (9) and (10), respectively:

$$Var\left\{\ln\left[Sa_{avg}\left(T_1...T_N\right)\right]\right\} = \frac{1}{N^2}\sum_{i=1}^{N}\sum_{j=1}^{N}\left[\rho_{\ln[Sa(T_i)],\ln[Sa(T_j)]}\sigma_{\ln[Sa(T_i)]}\sigma_{\ln[Sa(T_j)]}\right] \qquad (9)$$

$$\rho_{\ln\left[Sa_{avg}\left(T_1...T_N\right)\right],\ln[Sa(T_1)]} = \frac{\sum_{i=1}^{N}\rho_{\ln[Sa(T_i)],\ln[Sa(T_1)]}\sigma_{\ln[Sa(T_i)]}}{\sqrt{\sum_{i=1}^{N}\sum_{j=1}^{N}\left[\rho_{\ln[Sa(T_i)],\ln[Sa(T_j)]}\sigma_{\ln[Sa(T_i)]}\sigma_{\ln[Sa(T_j)]}\right]}} \qquad (10)$$

where the term $\rho\ln[Sa(T_i)]$, $\ln[Sa(T_j)]$ represents the correlation between spectral acceleration values at periods $T_i$ and $T_j$. The
correlation coefficients have been obtained by the authors of the present study (Rodríguez-Castellanos et al., 2019).

### 4.2 Values of $T_N$

Among the parameters that define the intensity measure $I_{Np}$, the geometric mean, $Sa_{avg}(T_1...T_N)$, has a crucial role when
computing the uniform hazard spectra (UHS). The $T_N$ value ($N$-th structural vibration period) takes into account the level of
nonlinearity developed by the structure. Bojórquez et al., (2008, 2011) recommend using $T_N$=2.0T1. Nevertheless, we consider
that there is no optimal period range for $Sa_{avg}(T_1...T_N)$ that meets the entire range of structural vibration periods; therefore, here
we propose that $T_N$ should depend on the structural vibration period, which is in agreement with Tsantaki et al., (2012, 2017).
It has been pointed out that the stiffer the structural system, the larger the period lengthening. Accordingly, for structures with
short vibration periods, we adopt $T_N$=2.0$T_1$, which agrees with recommendations made by Bianchini et al., (2009), Katsanos
and Sextos (2015), and Tsantaki et al., (2017), for relatively stiff structures, and assuming a ductility demand between 2 and
3.

At short-to-moderate vibration periods, the structural period lengthening diminishes somewhat linearly until it reaches a semi-
constant behavior (which is independent of the level of nonlinearity developed by the structure) [44]. In this sense, Di Sarno
and Amiri (2019) quantified the fundamental period lengthening of structures by the ratio of response spectra corresponding
to the lengthened and the elastic structural vibration period ($T_{in}/T_{el}$). They suggested dividing the response spectra into two
main regions: the first associated with short-to-moderate period structures, whose period shift ratio $T_{in}/T_{el}$ decreases with
increasing the elastic period; and the second region related to long-period structures, where the ratio period $T_{in}/T_{el}$ behaves
practically constant. Consequently, there must be a certain bound where the period shift ratio switches to remain constant. We
assume $T_N$=$T_s$ as the bound from which the lengthening of the structural vibration period remains almost constant.

For vibration periods longer than the soil dominant period, it is assumed $T_N$=1.25$T_1$, which is, on average, the period shift ratio-
value for structures with a short-to-moderate nonlinearity level, that is, with ductility ratio around 2 to 3 (Katsanos and Sextos,
2015; Di Sarno and Amiri, 2019).

Summarizing, we used in this study: $T_N$ =2.0$T_1$, for structural systems with short fundamental period; $T_N$=$T_s$ for those with
intermediate period; and $T_N$=1.25$T_1$ for systems with long fundamental period. It is possible to get a better approximation of





$T_N$ bounds, by means of a parametric study of the ratios of the equivalent period of SDOF degraded systems and that of the elastic systems ($T_{in}/T_{el}$), as a function of $T_{el}$, for a given ductility; such study can consider both ground motion characteristics
and structural properties (such as degrading stiffness ratio, pinching factor, accumulated damage factor, etc.), as it was done by Di Sarno and Amiri (2019). They proposed a mathematical expression for estimating the lengthening of the fundamental period as a function of the structural elastic period and the significant structural parameters, which is applicable to systems in sites classes D and C according to ASCE/SEI 7-10 (2010), with shear wave velocities $182.88 < V_{s30} < 365.76$ m/s and $365.76 < V_{S30} < 762$m/s, respectively. However, the $T_N$ bounds used here lead to reasonable results, as it is verified below.

**5 Probabilistic seismic hazard analysis using $I_{Np}$**

**5.1 Uniform hazard spectra corresponding to firm ground**

The uniform hazard spectra are computed, in first place, for the CU site, which is in firm ground. Fig. 3a shows the UHSs if only interplate, or alternatively, intraslab earthquakes occur. It also displays when both types of events are considered simultaneously (Total). Fig. 3b shows the total UHS of $Sa(T_1)$ and $I_{Np}$, both associated with a 250-year return period. It can be
seen that the spectra are quite similar; practically, they reach the same acceleration levels, and slight differences occur at long periods.

**5.2 Uniform hazard spectra corresponding to soft soil sites**

Estimating the seismic hazard at firm ground allows proceeding with a technique to assess the seismic hazard at soft soil sites. In this regard, Esteva (1970) presented a formulation in which through a known hazard curve at a reference site, it is feasible
to estimate a hazard curve at a recipient site. In this study, CU station is the reference site (we used it because, since 1964, it has recorded all the significant ground motions that have struck Mexico City). In addition, different studies have taken CU as a reference site (Ordaz et al., 1988; Reinoso and Ordaz, 1999; Singh et al., 1988).Therefore, it is viable to perform a hazard analysis for CU station and then to compute the annual rate of exceedance at other sites located in soft or medium soils, as follows:

$$v_Y(y) = \int_0^\infty v_X\left(\frac{y}{\tau}\right) f_\tau(\tau) d\tau = E_\tau\left(v_x\left(\frac{y}{z}\right)\right) \tag{11}$$

where:

$v_Y(y)$ is the mean annual rate of exceedance of a seismic IM, for the recipient site.

$v_x(y/\tau)$ is the mean annual rate of exceedance of a seismic IM for the reference site, divided by the variable $\tau$.

$\tau$ is the acceleration response spectral ratio ($Y/X$). It refers to the ratio between the response spectra corresponding to the
recipient and the reference sites.

$f\tau(\tau)$ is the pdf of $\tau$.



Figures 4a to 4f show the mean response spectral ratios for $Sa(T_1)$ (continuous lines) and $I_{Np}$ (interrupted lines), for one representative station located in each of the zones listed in Table 1. For this analysis, more than 1100 ground-motion records corresponding to the different recording stations were used. The stations are grouped depending on the soil dominant period

where these are located, as follows: Zone A: $T_s$<0.5s; Zone B: 0.5s<$T_s$<1.0s; Zone C: 1.0s<$T_s$<1.5s; Zone D: 1.5s<$T_s$<2.0s; Zone E: 2.0s< $T_s$<2.5s; and Zone F: 2.5s<$T_s$<3.0s. Fig. 5 shows the location of the recording stations used in this study.

Next, in order to compute the mean annual rate of exceedance of $Sa(T_1)$ and $I_{Np}$, the seismic hazard curves corresponding to CU station are coupled with the response spectral ratios, using Eq. (11). Figures 6a to 6f show the hazard curves ($\lambda$) of $Sa(T_1)$ and $I_{Np}$, for the same sites mentioned in Figures 4a to 4f, and for different vibration periods ($T_n$).

Then, having the mean rates of exceedance for each recording station site (see Table 1), the UHS are estimated for a given return interval. Figures 7a to 7f show the UHS of $Sa(T_1)$ and $I_{Np}$ for the same stations of figures 6a to 6f, for a 250-year return period. Fig. 7 shows that the spectral ordinates are comparable for both IMs, particularly at the firm ground (Zones A, B and C); however, at soft soil (Zones D, E and F), the spectral ordinates of $I_{Np}$ are notably higher than those corresponding to $Sa(T_1)$ at vibration periods shorter than the soil dominant period. In contrast, they are smaller at vibration periods longer than $T_s$. The

same can be appreciated for different sites of the city in the maps shown in Figs. 8a to 8d, which correspond to $Sa(T_1)$ (left side) and $I_{Np}$ (right side), for $T_1$=0.5s (up side) and $T_1$=1.0s (down side), for a return interval of $T_r$=250 years.

## 6 Degrading structural behavior effect

Once the uniform hazard spectra of $Sa(T_1)$ and $I_{Np}$ were estimated, the degrading structural behavior effect is evaluated by means of the ratio $I_{Np}/Sa(T_1)$. It represents the ratio of strength demands between a system with degrading, and the same system

with non-degrading structural behavior. The ratios are obtained for each station of the zones listed in Table 1. Figures 9a to 9f show the $I_{Np}/Sa(T_1)$ ratios (thin gray lines) as a function of the normalized periods $T_n/T_s$, for zones A to F, respectively.

Based on these ratios, it was proposed the following spectral modification function (SMF), which is a variation of that specified by MCSD (2008, 2015):

$$SMF = a + \frac{1}{b + c \left| d \frac{T_n}{T_s} - 1 \right|^e} \tag{12}$$

where the values of $a$, $b$, $c$ and $d$ are shown in Table 2. It is noticed that the values of the parameters depend on the type of soil where the structure is located; on the contrary, those in MCSD function are constant values; in addition, such function is restricted only to soft soils.

Figs. 9a to 9f show the equation proposed here (Eq. 12) (thick interrupted line), as well as the MCSD (2008, 2015) function (thick line). In the figures, the horizontal and vertical dotted lines, aligned at $I_{Np}/Sa(T_1)$=1 and $T_n/T_s$=1, delimitate

approximately the increase or decrease of the spectral amplification.



The figures show the following:

a) The highest $I_{Np}/Sa(T_1)$ ratios are reached for structures with vibration periods shorter than the dominant soil period

(approximately $T_s/2$), which indicates that the lateral strength demand for degrading systems is higher than the strength demand for non-degrading systems.

b) When the vibration period of the system is close to the soil dominant period ($T_n/T_s \approx 1$), the strength demands for degrading and non-degrading systems, are similar.

c) When $T_n/T_s > 1$, the demands of the degrading systems decrease with respect to those of the non-degrading systems. It means

that for structural vibration periods longer than $T_s$, the degrading behavior provides a beneficial effect.

d) It is noticed that for zone D (Fig. 9d), the MCSD function predicts spectral modification values which are similar to the function proposed in the present study (Eq.12). It happens because MCSD function was calibrated using ground motion data recorded a station located in that zone (SCT station in zone D); however, it does not happen the same for other soil conditions, especially for $T_n/T_s > 1$.

e) Equation (12) predicts values closer to unity at sites in zones A, B, and C (firm ground and transition soil), than at zones D, E and F, which means that the structural softening is not as significant as it is for zones D, E, and F. In this respect, several studies have observed that the degradation of the stiffness has little effect on the strength demands for structures located on firm sites (Akkar et al., 2004; Chenouda and Ashraf Ayoub, 2008; Chopra and Chintanapakdee, 2004). Moreover, it is noticed that at very short vibration period systems ($T_n/T_s < 0.1$), the SMF proposed here predicts amplification values very close to

unity, which is consistent for extremely stiff structures.

f) Finally, the reduction of strength demand according to Eq. (12) fits better the observed data (thin grey lines) for each type of soil (zones A to F) than that recommended by MCSD guidelines.

With the aim of verifying the validity of the proposed expression, Figures 10a and 10b compare the results of Eq. (12) with those obtained from time-history analysis of SDOF systems. The figures show the mean ratio of strength demands of degrading

and of non-degrading systems (elasto-plastic behavior) corresponding to a ductility value, $\mu_u$ (thin gray lines), using firm ground and soft soil records, respectively (Miranda and Ruiz-Garcia, 2002; Terán-Gilmore and Espinosa Johnson, 2008). The ground motions at firm ground (Fig.10a) correspond to synthetic accelerograms ($T_s$=1.0s) (Terán-Gilmore and Espinosa Johnson, 2008) and ground motions recorded in San Francisco bay area during 1989 the Loma Prieta earthquake ($T_s \approx 1.1$s) (Miranda and Ruiz-Garcia, 2002). In contrast, the ground motions at soft soil were recorded in the Lake Bed zone of Mexico

City ($T_s \approx 2.0$s) (Fig.10b).

Figures 10a and 10b also include the $I_{Np}/Sa(T_1)$ ratios, corresponding to the stations D11 and C3, estimated from the uniform hazard spectra normalization (thick red dotted lines). It can be observed that the $I_{Np}/Sa(T_1)$ ratio agrees with results of Miranda and Ruiz-Garcia (2002), and of Terán-Gilmore and Espinoza-Johnson (2008). The figures also show that the function given by Eq. (12) is in agreement with both the observed data obtained from the time-history analyses and the $I_{Np}/Sa(T_1)$ ratio

calculated from the study based on seismic hazard analyses.



# 7 Conclusions

A methodology based on probabilistic seismic hazard analysis is proposed to evaluate the effect of degrading behavior on the strength demands of SDOF systems. For this aim, there are obtained uniform hazard spectra for two alternative intensity measures: $I_{Np}$ and $Sa(T_1)$, which represent the response of degrading and non-degrading systems, respectively; so, the ratio of the hazard spectra $I_{Np}/Sa(T_1)$ characterizes the strength demands of systems with degrading behavior to those of systems with non-degrading behavior. Based on the $I_{Np}/Sa(T_1)$ ratios, which correspond to systems located at different sites, grouped in different seismic zones (depending on the type of soil where the structures are located), a mathematical expression is proposed. The methodology is applied here to structural systems located in Mexico City, but it can be applied to any seismic region of the world.

From the study the following is concluded;

1. For structures with vibration periods shorter than the dominant soil period ($T_n/T_s<1$), degrading systems exhibit strength demands up to 30% higher, than systems with non-degrading behavior.

2. For structures with vibration periods close to the soil dominant period ($T_n/T_s\approx1$) the strength demands for degrading and non-degrading systems, are similar.

3. For systems with vibration periods longer than the soil dominant period ($T_n/T_s>1$), the strength demands for structures with degrading behavior are lower, approximately 5% to 20%, than structures with non-degrading behavior. That reduction highly depends on the soil dominant period at the site, and it is larger for systems with longer soil dominant periods. For these cases, the structural degrading behavior produces a beneficial effect, reducing the lateral strength requirement of the structures.

4. A strength modification factor was proposed (Eq. 12). The expression was fitted according to the spectral ratios $I_{Np}/Sa(T_1)$ corresponding to different soil conditions. The value of the parameters included in the equation depends on the type of soil where the structure is located.

5. The expression proposed (Eq. 12) is a useful tool for simplified nonlinear modal analyses, to incorporate explicitly the effect of degrading behavior according to the type of soil where the structure is located. It was verified that the mathematical expression proposed leads to results that are comparable to those obtained from time history analyses of SDOF systems located in soft soil.

6. It has been proposed to the Authorities, to incorporate the expression proposed here in the next version of Mexico City Building Code.

7. In addition, the study presents a methodology to elaborate seismic hazard maps in terms of the intensity measure $I_{Np}$. Based on that methodology, it is presented the first seismic hazard map of Mexico City, in terms of $I_{Np}$.




**Author contribution**

**Sonia E. Ruiz:** Conceptualization, Methodology, Writing-Original draft preparation. **Ali Rodríguez-Castellanos:** Formal
analysis, Software, Writing-Original draft preparation. **Edén Bojórquez:** Conceptualization, Writing-Original draft
preparation. **Miguel A. Orellana:** Resources, Software. **Alfredo Reyes-Salazar:** Resources, Visualization.

**Competing interests**

We wish to confirm that there are no known conflicts of interest associated with this publication and there has been no
significant financial support for this work that could have influenced its outcome.
We confirm that the manuscript has been read and approved by all named authors and that there are no other persons who
satisfied the criteria for authorship but are not listed. We further confirm that the order of authors listed in the manuscript has
been approved by all of us. We confirm that we have given due consideration to the protection of intellectual property
associated with this work and that there are no impediments to publication, including the timing of publication, with respect
to intellectual property. In so doing we confirm that we have followed the regulations of our institution concerning intellectual
property.
We understand that the first and the second Corresponding Author are the contacts for the Editorial process (including Editorial
Manager and direct communications with the office). They are responsible for communicating with the other authors about
progress, submissions of revisions and final approval of proofs. We confirm that we have provided current, correct email
addresses which are accessible by the Corresponding Authors and which have been configured to accept email from
SruizG@iingen.unam.mx and eden@uas.edu.mx.

**Acknowledgements**

Thanks are given to DGAPA-UNAM (project PAPIIT IN100320) and to Instituto para la Seguridad de las Construcciones de
la Ciudad de Mexico, for their support. The second and fourth authors acknowledge the scholarship given by Consejo Nacional
de Ciencia y Tecnología (CONACyT) during their postgraduate studies. The authors wish to thank CIRES for having provided
the seismic records used in this study.

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



**Table 1. Zones of Mexico City grouped in accordance with the soil dominant period.**

| Zones | Station | $T_s$(s) | Station | $T_s$(s) | Average $T_s$(s) |
|---|---|---|---|---|---|
| Zone A | A1 | 0.5 | A4 | 0.4 | |
| | A2 | 0.5 | A5 | 0.5 | 0.5 |
| | A3 | 0.5 | A6 | 0.5 | |
| Zone B | B1 | 0.9 | B6 | 0.8 | |
| | B2 | 0.9 | B7 | 0.8 | |
| | B3 | 0.7 | B8 | 0.7 | 0.75 |
| | B4 | 0.6 | B9 | 1.1 | |
| | B5 | 0.7 | B10 | 0.8 | |
| Zone C | C1 | 1.4 | C4 | 1.3 | |
| | C2 | 1.4 | C5 | 1.3 | 1.3 |
| | C3 | 1.4 | C6 | 1.2 | |
| Zone D | D1 | 1.8 | D7 | 2 | |
| | D2 | 1.7 | D8 | 2 | |
| | D3 | 1.7 | D9 | 1.8 | |
| | D4 | 2.1 | D10 | 2.2 | 1.9 |
| | D5 | 2 | D11 | 1.7 | |
| | D6 | 2 | D12 | 1.8 | |
| Zone E | E1 | 2.4 | E4 | 2 | |
| | E2 | 2.3 | E5 | 2.1 | 2.3 |
| | E3 | 2.2 | E6 | 2.3 | |
| Zone F | F1 | 2.7 | F4 | 2.6 | |
| | F2 | 2.5 | F5 | 2.5 | 2.7 |
| | F3 | 2.7 | F6 | 2.9 | |

**Table 2. Numerical coefficients for SMF expression (Eq. 12).**

| Zone | $a$ | $b$ | $c$ | $d$ | $e$ |
|---|---|---|---|---|---|
| A | 1.0 | 3.5 | 12.0 | 2.0 | 3.0 |
| B | 0.9 | 3.0 | 8.5 | 2.0 | 3.5 |
| C | 0.9 | 2.5 | 5.0 | 2.0 | 4.0 |
| D | 0.8 | 2.0 | 3.0 | 2.0 | 4.5 |
| E | 0.8 | 1.9 | 2.1 | 2.3 | 4.9 |
| F | 0.7 | 1.7 | 1.8 | 2.1 | 5.5 |







Figure 1: Block diagram of the proposed methodology.







| Zone | p | q |
|------|------|------|
| SUB1 | 0.55 | 2.39 |
| SUB2 | 0.75 | 3.95 |
| SUB3 | 0.77 | 4.20 |
| SUB4 | 0.75 | 4.07 |

| Zone | p | q |
|------|------|------|
| IN1 | 0.80 | 3.50 |
| IN2 | 0.83 | 4.35 |
| IN3 | 0.82 | 3.77 |

| Zone | $v_7$ | Zone | $v_7$ |
|------|--------|------|--------|
| C1 | 0.0369 | C8 | 0.0112 |
| C2 | 0.0334 | C9 | 0.0223 |
| C3 | 0.0279 | C10 | 0.0156 |
| C4 | 0.0190 | C11 | 0.0335 |
| C5 | 0.0134 | C12 | 0.0178 |
| C6 | 0.0112 | C13 | 0.0167 |
| C7 | 0.0290 | C14 | 0.0456 |

$E_{Mw}=7.5$
$\sigma_{Mw}=0.30$

**Figure 2: a) Interplate seismicity regions, b) intraslab seismicity regions, and c) characteristic seismicity regions.**








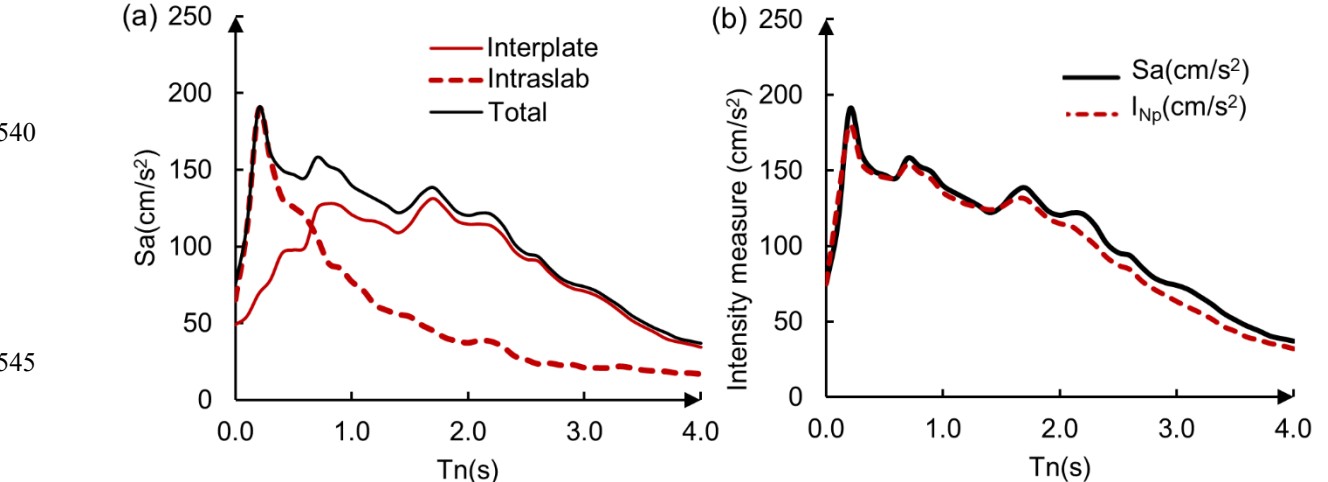

**Figure 3. (a) Uniform hazard spectra for CU, and (b) uniform hazard spectra of $Sa(T_1)$ and $I_{Np}$, for CU (250 year-return period).**












**Figure 4: Mean response spectral ratios for $Sa(T_1)$ and $I_{Np}$ corresponding to one representative station of each zone listed in Table 1.**




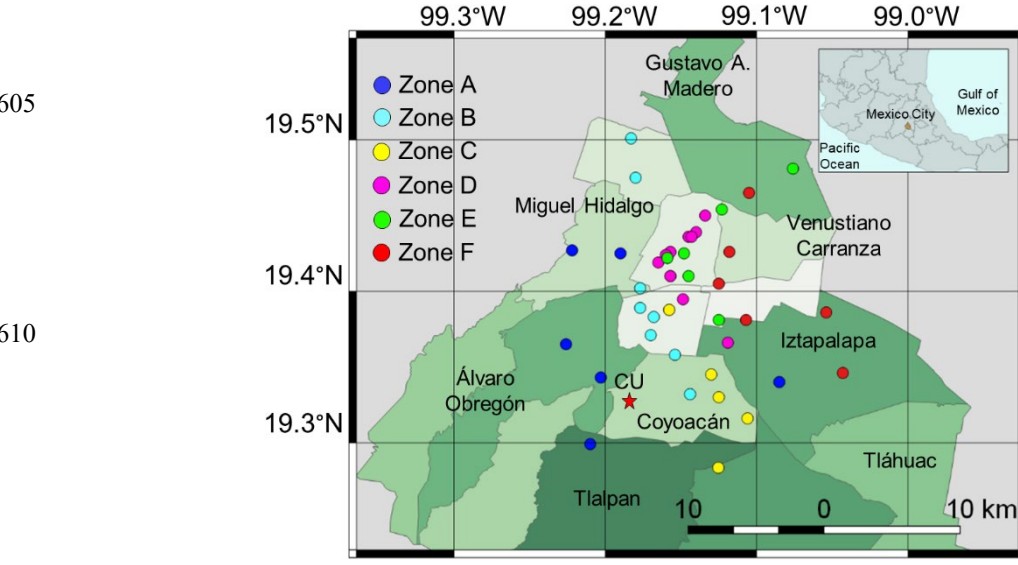

**Figure 5: Locations of seismic recording stations in Mexico City (see Table 1).**







**Figure 6: Mean annual rate of exceedance ($\lambda$) of $Sa(T_1)$ and $I_{Np}$, for different vibration periods, corresponding to one representative station of each zone listed in Table 1.**









**Figure 7: Uniform hazard spectra of *Sa(T₁)* and *I_Np*, corresponding to one representative station of each zone listed in Table 1, considering 250 year-return interval.**


**Figure 8: Intensity maps corresponding to *Sa*(*T₁*) (left side) and *I_Np* (right side), for *T₁*=0.5s (up side) and *T₁*=1.0s (down side), for 250-year return interval.**






**Figure 9: Spectral ratios between the uniform hazard spectra of $I_{Np}$ and $Sa(T_1)$ ($I_{Np}/Sa(T_1)$), for the recording stations, corresponding to six zones in Mexico City (see Table 1).**




745

760

765

**Figure 10: Mean ratios of strength demands of degrading and of non-degrading systems corresponding to a) firm ground (zone C), and b) soft soil (zone D) of Mexico City.**