# Peer review of "Reliability-based strength modification factor for seismic design spectra considering structural degradation"

_Natural Hazards and Earth System Sciences, 2020_

## Referee Comment (RC1) · Anonymous Referee #1 · 25 Aug 2020

This paper describes an approach to obtain strength modification factors of design spectra for degrading structures, based on probabilistic seismic hazard analyses, to avoid running nonlinear time history analyses for SDOFs. The paper is well written, well structured, and comprehensive. The presented methodology is clear, and the results and figures support the findings of the Authors.

It is this Reviewer's opinion that the paper is suitable for publication in NHESS, subject to the following minor revisions: 1. An important aspect when calculating the proper modification of response spectra, besides the structural degradation and the soil contribution, is the soil-structure interaction, which might further modify the response spec-

trum. NIST2012 describes a procedure of accounting for SSI in the response spectra calculation. This modification is more pronounced for stiff structures resting on soft soil. Can you please comment on how you believe that SSI could further modify the response spectra that you calculated? Also, a reference to this matter could be made in the Introduction section. 2. In Section 3.4, please provide some indicative properties for the soil at the CU station, in order to support your statement that it is firm ground. 3. Please change 2.4 to 3.4 in Line 177. 4. Please provide some comments on why you chose $T_N = T_s$ as the bound from which the lengthening of T remains constant. 5. Figures 4, 5, and 6 lack commentary in the text (simple reference is made). Please add your comments and discussion to those figures within the text, as the presented results are of interest to the reader. 6. In Figure 7, please add, whenever possible, a percent of the difference between $S_a$ and $I_{Np}$. This will help the reader to quantify the effect of structural degradation. 7. In Lines 263-264, please rephrase "interrupted" with "dashed" and add "thick solid line".

---

## Author Comment (AC1) · 1 Sep 2020

Dear Referee #1:

We appreciate the time and effort that you have dedicated to providing your valuable observations on our manuscript. We especially appreciate your comment that says that our results are interesting and are suitable for publication in NHESS journal. We have incorporated all the changes suggested, and have highlighted the changes in the manuscript.

Comments from Referee # 1

[Figure]

Comment 1: An important aspect when calculating the proper modification of response spectra, besides the structural degradation and the soil contribution, is the soil-structure interaction, which might further modify the response spectrum. NIST2012 describes a procedure of accounting for SSI in the response spectra calculation. This modification is more pronounced for stiff structures resting on soft soil. Can you please comment on how you believe that SSI could further modify the response spectra that you calculated? Also, a reference to this matter could be made in the Introduction section.

Reply 1: The SSI effect is an important topic in earthquake engineering, and particularly for structures located at soft soil of Mexico City. Although in the present study the SSI was not considered, we have modified the section "Introduction" according with the suggestion of the reviewer in order to illustrate the effect of SSI for earthquake resistant design. The following paragraph was added to the new version of the manuscript:

"It is important to say that in the present work, soil structure interaction (SSI) was not taken into account to compute the strength modification factors for seismic design spectra; however, notice that the effect of SSI is more important for stiff structures located on soft soil, in such a way that for this type of structures, the ordinates of the response spectra tend to increase while the opposite occurs for flexible structures (Avilés and Pérez-Rocha, 2007). The results obtained in the present work could be modified to include the effect of SSI via the current Mexico City Building Code (MCBC, 2017) which provide recommendation about this issue".

MCBC: Normas Técnicas Complementarias para Diseño por Sismo de la ciudad de México, CDMX, Mexico., 2017.

Avilés, J y L E Pérez-Rocha (2007), "Damage analysis of structures on elastic foundation", Journal of Structural Engineering, ASCE, Vol. 133(10), pp.1453-1461. DOI:10.1061/(ASCE)0733-9445(2007)133:10(1453)

Comment 2: In Section 3.4, please provide some indicative properties for the soil at

the CU station, in order to support your statement that it is firm ground.

Reply 2: We agree with the referee; for this reason, lines 137 to 138 in Section 3.4 now read as follows:

"They were developed using accelerometric data recorded in Ciudad Universitaria station (CU), which is located at the hill zone (firm ground) of Mexico City, basically conformed by a surface layer of lava flows and volcanic tuffs with a shear wave velocity in the upper 30 m of 750 m/s (Ordaz and Singh, 1992; Singh et al., 2018)."

Comment 3: Please change 2.4 to 3.4 in Line 177.

Reply 3: Thanks. We have corrected line 184 in the new version of the manuscript.

Comment 4: Please provide some comments on why you chose TN=Ts as the bound from which the lengthening of T remains constant.

Reply 4: We agree with the referee; for that reason, in Section 4.2, lines 207 to 211, we have included our reasoning to propose TN=Ts as the bound from which the lengthening of the structural vibration period remains almost constant. The paragraph added to the manuscript says:

"They suggested dividing the response spectra into two main regions: the first associated with short-to-moderate period structures, whose period shift ratio Tin/Tel decreases with increasing the elastic period; and the second region related to long-period structures, where the ratio period Tin/Tel behaves practically constant. Consequently, there must be a certain bound where the period shift ratio switches to remain constant; therefore, we propose TN = Ts as that bound from which the lengthening of the structural vibration period remains almost constant. In this context, Miranda and Ruiz-Garcia (2002, 2003), and independently, Terán-Gilmore and Espinoza (2008), found that strength demands between degrading and non-degrading systems are similar when the structural period and dominant soil period are comparable, which means that the mean ratio value should be approximate to one when Tn ≈ Ts. "

Comment 5: Figures 4, 5, and 6 lack commentary in the text (simple reference is made). Please add your comments and discussion to those figures within the text, as the presented results are of interest to the reader.

Reply 5: We agree with your suggestion. We have added some comments about Figures 4, 5, and 6. The comments have been incorporated in lines 245 to 264 in the new version of the manuscript. The paragraphs added to the manuscript say:

"Therefore, to evaluate the previous function, firstly, the spectral ratios are estimated, and then are coupled with the seismic hazard curves via Eq. (11). In this respect, Figures 4a to 4f show the mean response of the spectral ratios for Sa(T1) (solid line) and INp (dashed line) for one representative station located in each of the zones listed in Table 1. In this sense, the spectral ratios roughly represent the spectral amplification of soft soil with respect to firm ground. It is observed how the peak values shift towards increasingly longer periods, which, approximately, match with the dominant soil period (see Table 1). For this analysis, more than 1100 ground-motion records corresponding to the different recording stations were used. The stations are grouped depending on the soil dominant period where these are located, as follows: Zone A: Ts<0.5s; Zone B: 0.5s<Ts<1.0s; Zone C: 1.0s<Ts<1.5s; Zone D: 1.5s<Ts<2.0s; Zone E: 2.0s< Ts<2.5s; and Zone F: 2.5s<Ts<3.0s. Additionally, Fig. 5 shows the location of the recording stations in Mexico City, which are represented with circles of different colours associated with each of the proposed zones (see Table 1). Next, in order to compute the mean annual rate of exceedance of Sa(T1) and INp, the seismic hazard curves corresponding to CU station are coupled with the response spectral ratios, using Eq. (11). Figures 6a to 6f show the hazard curves ($\lambda$) of Sa(T1) and INp, associated with different vibration periods, corresponding to CU and the same recording stations of Figures 4a to 4f. In the first place, as expected, the rates of exceedance for all the recording stations analyzed are higher than the corresponding ones of CU (up and down, respectively). Additionally, concerning the CU site, the hazard curves for both intensity measures INp and Sa(T1) are very similar, and differences are barely visible

at long return periods. Now, for the rest of the recording stations, Figs. 4e and 4f show noticeable variations between exceedance rates of Sa(T1) and INp; nevertheless, Figs. 4e and 4f display almost no contrast between the rates of exceedance of the two intensity measures. The previous is relative, because to fully characterize the variations between exceedance rates of Sa(T1) and INp, a wide range of periods needs to be covered; for this reason, we estimate the UHS in the following."

Comment 6: In Figure 7, please add, whenever possible, a percent of the difference between Sa and INp. This will help the reader to quantify the effect of structural degradation.

Reply 6: We have added some comments about the percentage difference between Sa and INp, in Section 5.2, lines 267 to 270. The paragraph added to the manuscript says:

"It is observed that, at vibration periods shorter than the dominant soil period, the spectral ordinates corresponding to firm ground (Zones A, B, and C) are comparable for both IMs. However, at soft soil (Zones D, E, and F), the spectral ordinates of INp are notably higher than those of Sa(T1) (up to 30%). In contrast, at vibration periods longer than Ts, they are smaller than those corresponding to Sa(T1) (5% to 20%, depending on the soil type)."

Comment 7: In Lines 263-264, please rephrase "interrupted" with "dashed" and add "thick solid line".

Reply 7: We agree with your suggestion. We have corrected it in lines 284 and 285 of the new version of the manuscript.
* * *

---

## Referee Comment (RC2) · Anonymous Referee #2 · 4 Sep 2020

The submitted manuscript concerns a new approach for the definition of the strength modification factor of design spectra to be used in the seismic design of degrading structures. In the opinion of this reviewer the considered topic is very interesting from scientific point of view, the paper is well written and the subject falls within the scope of the "Natural Hazards and Earth System Sciences" journal. Summarizing, I would recommend the paper for publication in NHESS considering only minor revision, as indicated in the following comment. The rate of variation of the vibration periods of a structure, in the passage from the undamaged to the damaged state, strongly depends on the structural type, on the interaction of the structural elements with the non-structural ones, as well as on the soil-structure interaction. Could the authors comment

on whether the above parameters may have an effect on the proposed procedure, also considering the different design limit states furnished by current codes?
* * *

---

## Author Comment (AC2) · 7 Sep 2020

Referee #2 The submitted manuscript concerns a new approach for the definition of the strength modification factor of design spectra to be used in the seismic design of degrading structures. In the opinion of this reviewer the considered topic is very interesting from scientific point of view, the paper is well written and the subject falls within the scope of the "Natural Hazards and Earth System Sciences" journal. Summarizing, I would recommend the paper for publication in NHESS considering only minor revision, as indicated in the following comment.

Authors: Dear Referee # 2 We appreciate the time and effort that you have dedicated

to provide your valuable observations on our manuscript, especially your comment regarding to the recommendation of our paper for publication in NHESS considering only minor revision.

Referee # 2 The rate of variation of the vibration periods of a structure, in the passage from the undamaged to the damaged state, strongly depends on the structural type, on the interaction of the structural elements with the nonstructural ones, as well as on the soil-structure interaction. Could the authors comment on whether the above parameters may have an effect on the proposed procedure, also considering the different design limit states furnished by current codes?

Authors: Thank you very much for this observation, we completely agree with the reviewer. The rate of variation of the vibration periods of a structure from the undamaged to the damaged state strongly depends of several parameters, and this is crucial to consider different design limit states. It is important to say that although the procedure is not affected by those parameters, the variation of the structural period could be taken into account considering different values of TN; however, the definition of this value accounting for the design limit state, structural type, interaction of the structural elements with the nonstructural ones, as well as the soil-structure interaction requires the study of specific structural systems such as: steel structures, reinforced concrete moment resisting frames, masonry structures, buildings with braces, posttensioned and based isolated structures, among others, which is out of the scope of the present study. Notice that currently the group of research on this subject of the Universidad Autonoma de Sinaloa and of the Universidad Nacional Autonoma de Mexico are working in order to develop optimal values of TN for different types of structural systems and accounting for the parameters indicated by the reviewer.

---

## Author Response (AR1)

**Authors´ reply to Referee #1 comments to the manuscript** *"Reliability-based strength modification factor for seismic design spectra considering structural degradation"*. **Paper ID: https://doi.org/10.5194/nhess-2020-116-RC1, 2020**

Dear Referee #1:

We appreciate the time and effort that you have dedicated to providing your valuable observations on our manuscript. We especially appreciate your comment that says that our results are interesting and are suitable for publication in NHESS journal. We have incorporated all the changes suggested, and have highlighted the changes in the manuscript using yellow color.

**Comments from Referee # 1**

**Comment 1:** An important aspect when calculating the proper modification of response spectra, besides the structural degradation and the soil contribution, is the soil-structure interaction, which might further modify the response spectrum. NIST2012 describes a procedure of accounting for SSI in the response spectra calculation. This modification is more pronounced for stiff structures resting on soft soil. Can you please comment on how you believe that SSI could further modify the response spectra that you calculated? Also, a reference to this matter could be made in the Introduction section.

**Reply 1:** The SSI effect is an important topic in earthquake engineering, and particularly for structures located at soft soil of Mexico City. Although in the present study the SSI was not considered, we have modified the section "Introduction" according with the suggestion of the reviewer in order to illustrate the effect of SSI for earthquake resistant design. The following paragraph was added to the new version of the manuscript:

*"It is important to say that in the present work, soil structure interaction (SSI) was not taken into account to compute the strength modification factors for seismic design spectra; however, notice that the effect of SSI is more important for stiff structures located on soft soil, in such a way that for this type of structures, the ordinates of the response spectra tend to increase while the opposite occurs for flexible structures (Avilés and Pérez-Rocha, 2007). The results obtained in the present work could be modified to include the effect of SSI via the current Mexico City Building Code (MCBC, 2017) which provide recommendation about this issue".*

*MCBC: Normas Técnicas Complementarias para Diseño por Sismo de la ciudad de México, CDMX, Mexico., 2017.*

*Avilés, J y L E Pérez-Rocha (2007), "Damage analysis of structures on elastic foundation", Journal of Structural Engineering, ASCE, Vol. 133(10), pp.1453-1461. DOI:10.1061/(ASCE)0733-9445(2007)133:10(1453)*

**Comment 2:** In Section 3.4, please provide some indicative properties for the soil at the CU station, in order to support your statement that it is firm ground.

**Reply 2:** We agree with the referee; for this reason, lines 143 to 144 in Section 3.4 now read as follows:

*"They were developed using accelerometric data recorded in Ciudad Universitaria station (CU), which is located at the hill zone (firm ground) of Mexico City, basically conformed by a surface layer of lava flows and volcanic tuffs with a shear wave velocity in the upper 30 m of 750 m/s (Ordaz and Singh, 1992; Singh et al., 2018)."*

**Comment 3:** Please change 2.4 to 3.4 in Line 177.

**Reply 3:** Thanks. We have corrected line 190 in the new version of the manuscript.

**Comment 4:** Please provide some comments on why you chose $T_N = T_s$ as the bound from which the lengthening of T remains constant.

**Reply 4:** We agree with the referee; for that reason, in Section 4.2, lines 210 to 217, we have included our reasoning to propose $T_N = T_s$ as the bound from which the lengthening of the structural vibration period remains almost constant. The paragraph added to the manuscript says:

*"They suggested dividing the response spectra into two main regions: the first associated with short-to-moderate period structures, whose period shift ratio $T_{in}/T_{el}$ decreases with increasing the elastic period; and the second region related to long-period structures, where the ratio period $T_{in}/T_{el}$ behaves practically constant. Consequently, there must be a certain bound where the period shift ratio switches to remain constant; therefore, we propose $T_N = T_s$ as that bound from which the lengthening of the structural vibration period remains almost constant. In this context, Miranda and Ruiz-Garcia (2002, 2003), and independently, Terán-Gilmore and Espinoza (2008), found that strength demands between degrading and non-degrading systems are similar when the structural period and dominant soil period are comparable, which means that the mean ratio value should be approximate to one when $T_n \approx T_s$."*

**Comment 5:** Figures 4, 5, and 6 lack commentary in the text (simple reference is made). Please add your comments and discussion to those figures within the text, as the presented results are of interest to the reader.

**Reply 5:** We agree with your suggestion. We have added some comments about Figures 4, 5, and 6. The comments have been incorporated in lines 251 to 270 in the new version of the manuscript. The paragraphs added to the manuscript say:

*"Therefore, to evaluate the previous function, firstly, the spectral ratios are estimated, and then are coupled with the seismic hazard curves via Eq. (11). In this respect, Figures 4a to 4f show the mean response of the spectral ratios for $Sa(T_1)$ (solid line) and $I_{Np}$ (dashed line) for one representative station located in each of the zones listed in Table 1. In this sense, the spectral ratios roughly represent the spectral amplification of soft soil with respect to firm*

*ground. It is observed how the peak values shift towards increasingly longer periods, which, approximately, match with the dominant soil period (see Table 1). For this analysis, more than 1100 ground-motion records corresponding to the different recording stations were used. The stations are grouped depending on the soil dominant period where these are located, as follows: Zone A: $T_s<0.5s$; Zone B: $0.5s<T_s<1.0s$; Zone C: $1.0s<T_s<1.5s$; Zone D: $1.5s<T_s<2.0s$; Zone E: $2.0s< T_s<2.5s$; and Zone F: $2.5s<T_s<3.0s$. Additionally, Fig. 5 shows the location of the recording stations in Mexico City, which are represented with circles of different colours associated with each of the proposed zones (see Table 1).*

*Next, in order to compute the mean annual rate of exceedance of $Sa(T_1)$ and $I_{Np}$, the seismic hazard curves corresponding to CU station are coupled with the response spectral ratios, using Eq. (11). Figures 6a to 6f show the hazard curves ($\lambda$) of $Sa(T_1)$ and $I_{Np}$, associated with different vibration periods, corresponding to CU and the same recording stations of Figures 4a to 4f. In the first place, as expected, the rates of exceedance for all the recording stations analyzed are higher than the corresponding ones of CU (up and down, respectively). Additionally, concerning the CU site, the hazard curves for both intensity measures $I_{Np}$ and $Sa(T_1)$ are very similar, and differences are barely visible at long return periods. Now, for the rest of the recording stations, Figs. 4e and 4f show noticeable variations between exceedance rates of $Sa(T_1)$ and $I_{Np}$; nevertheless, Figs. 4e and 4f display almost no contrast between the rates of exceedance of the two intensity measures. The previous is relative, because to fully characterize the variations between exceedance rates of $Sa(T_1)$ and $I_{Np}$, a wide range of periods needs to be covered; for this reason, we estimate the UHS in the following."*

**Comment 6:** In Figure 7, please add, whenever possible, a percent of the difference between Sa and $I_{Np}$. This will help the reader to quantify the effect of structural degradation.
**Reply 6:** We have added some comments about the percentage difference between Sa and $I_{Np}$, in Section 5.2, lines 273 to 276. The paragraph added to the manuscript says:

*"It is observed that, at vibration periods shorter than the dominant soil period, the spectral ordinates corresponding to firm ground (Zones A, B, and C) are comparable for both IMs. However, at soft soil (Zones D, E, and F), the spectral ordinates of $I_{Np}$ are notably higher than those of $Sa(T_1)$ (up to 30%). In contrast, at vibration periods longer than $T_s$, they are smaller than those corresponding to $Sa(T_1)$ (5% to 20%, depending on the soil type)."*

**Comment 7:** In Lines 263-264, please rephrase "interrupted" with "dashed" and add "thick solid line".
**Reply 7:** We agree with your suggestion. We have corrected it in lines 290 and 291 of the new version of the manuscript.

**Authors´ reply to Referee #2 comments to the manuscript** *"Reliability-based strength modification factor for seismic design spectra considering structural degradation".* **Paper ID: https://doi.org/10.5194/nhess-2020-116-RC1, 2020**

Dear Referee #2:

We appreciate the time and effort that you have dedicated to provide your valuable observations on our manuscript, especially your comment regarding to the recommendation of our paper for publication in NHESS considering only minor revision. We have incorporated all the changes suggested, and have ==highlighted the changes in the manuscript using green color.==

**Comments from Referee # 2**

**Comment 1:** The rate of variation of the vibration periods of a structure, in the passage from the undamaged to the damaged state, strongly depends on the structural type, on the interaction of the structural elements with the nonstructural ones, as well as on the soil-structure interaction. Could the authors comment on whether the above parameters may have an effect on the proposed procedure, also considering the different design limit states furnished by current codes?

**Reply 1:** Thank you very much for this observation, we completely agree with the reviewer. The rate of variation of the vibration periods of a structure from the undamaged to the damaged state strongly depends of several parameters, and this is crucial to consider different design limit states. It is important to say that although the procedure is not affected by these parameters, the variation of the structural period could be taking into account considering different values of $T_N$; however, the definition of this value accounting for the design limit state, structural type, interaction of the structural elements with the nonstructural ones, as well as the soil-structure interaction requires the study of specific structural systems such as: reinforced concrete, moment resisting steel frames, masonry, structures with eccentrically, buckling restrained braces, posttensioned, based isolators among others, which is out of the scope of the present study. Notice that currently the group of research of the Universidad Autónoma de Sinaloa and Universidad Nacional Autónoma de Mexico are working in order to develop optimal values of $T_N$ for different types of structural systems and accounting for the parameters indicated by the reviewer. Therefore, we have modified the section "Introduction" according with the observation of the reviewer in order to consider this subject. The following paragraph was added to the new version of the manuscript:

[revised manuscript text omitted]